# Therapeutic Penetrating Keratoplasty in a Case of Corneal Perforation Caused by *Colletotrichum gloeosporioides* Infection

**DOI:** 10.3390/pathogens11050526

**Published:** 2022-04-29

**Authors:** Kazuki Imai, Takayoshi Sumioka, Hiroki Iwanishi, Yukihisa Takada, Shin’ichi Murata, Ryuta Iwamoto, Yuka Okada, Shizuya Saika

**Affiliations:** 1Department of Ophthalmology, School of Medicine, Wakayama Medical University, Wakayama 641-0012, Japan; imai-k@wakayama-med.ac.jp (K.I.); iwanishi@wakayama-med.ac.jp (H.I.); ytakada@wakayama-med.ac.jp (Y.T.); yokada@wakayama-med.ac.jp (Y.O.); shizuya@wakayama-med.ac.jp (S.S.); 2Department of Clinical Pathology, School of Medicine, Wakayama Medical University, Wakayama 641-0012, Japan; smurata@wakayama-med.ac.jp (S.M.); riwamoto@wakayama-med.ac.jp (R.I.)

**Keywords:** *Colletotrichum gloeosporioides*, corneal ulcer, glucocorticoid, therapeutic penetrating keratoplasty

## Abstract

**Background**: Corneal infection of *Colletotrichum gloeosporioides* is uncommon and usually limited to the anterior stroma. However, we observed a case of corneal stromal perforation caused by this fungus under a compromised condition. **Case**: A 73-year-old woman consulted us with a severe corneal ulceration. She was a tangerine orange farmer who suffered from rheumatoid arthritis for more than ten years. Before consultation with us, she received pterygium excision in her right eye. She then developed a corneal ulceration and received topical glucocorticoid therapy upon diagnosis of rheumatoid arthritis-related stromal ulcer in the eye. At the first consultation with us, a corneal ulceration was observed in the inferotemporal area of her right cornea. Biological examination detected a filamentous fungus, *Colletotrichum gloeosporioides*. Topical and systemic antifungal treatments were not significantly effective. Fourteen days after consultation, the lesion grew worse, leading to stromal perforation, which was treated by therapeutic penetrating keratoplasty using a preserved corneal button. **Conclusions**: Topical glucocorticoid could accelerate the growth of *Colletotrichum gloeosporioides* before diagnosis, even though the primary cause of corneal ulceration development might be rheumatoid arthritis.

## 1. Introduction

*Colletotrichum* spp. is a family of approximately 40 filamentous fungi seen in the tropics and subtropics. They usually grow in the plant body of many crops. However, 7 *Colletotrichum* spp. can reportedly cause infections in humans and 6 of those were reported as pathogens involved with corneal infection [1,2,3,4,5]. Corneal infection by *Colletotrichum* spp. is usually related to previous traumatic or surgical injuries. Additionally, *Colletotrichum gloeosporioides* is the major *Colletotrichum* spp. detected in corneal infection [3,6,7,8,9,10]. Anti-fungal drugs, such as voriconazole, fluconazole and natamycin, are reportedly effective for treatment of corneal infection [2,3,4,9,10].

Here, we report a case of severe corneal infection by *Colletotrichum gloeosporioides* that was treated by using therapeutic penetrating keratoplasty. Furthermore, topical glucocorticoid administration might accelerate the severity of the disease before detection of the pathogen.

## 2. Clinical Course

### 2.1. Patinet

A 73-year-old woman consulted us with a severe corneal ulceration. She was a tangerine orange farmer who suffered from rheumatoid arthritis for more than ten years. Before consultation with us, she received pterygium excision in her right eye. She then developed a corneal ulceration and received topical glucocorticoid therapy upon diagnosis of rheumatoid arthritis-related stromal ulcer in her eye.

At our first consultation, severe hyperemia, stromal infiltration (arrowheads), epithelial defect, and hypopyon with niveau were observed (Figure 1a). Fluorescein staining revealed the area of the epithelial defect (Figure 1b). Swab samples and stromal biopsy specimens for bacteriological examination were obtained. Before consultation with us, the patient received topical administration of 0.1% levofloxacin and 0.1% betamethasone sodium phosphate for one month upon diagnosis of peripheral ulcerative keratitis. After consultation with us, topical dexamethasone was discontinued upon possible diagnosis of of infection by some organism(s). In our inpatient clinic, in addition to topical levofloxacin (four times/day) and topical Cefmenoxime Hydrochloride (four times/day), the patient received an infusion mixture of Ampicillin Sodium (4 g) and Sulbactam Sodium (2 g) per day until we received the biological culture test results. We followed the treatment guidelines set by The Japanese Ophthalmological Society and chose systemic antibiotics administration. The biological culture using Sabouraud’s agar detected a filamentous fungus (Figure 2a); therefore, anti-fungal treatment (topical 1% voriconazole every one hour, 5% natamycin every four hours, and systemic voriconazole of 800 mg/day) was initiated. Further fungal examination detected *Colletotrichum gloeosporioides* (Figure 2b). Despite the treatment, the corneal lesion gradually enlarged and exhibited fibrin reaction alongside hypopyon in the anterior chamber (Figure 3a). Fourteen days after beginning consultation with us, the lesion got worse, leading to stromal perforation, which was not improved by wearing a medical soft contact lens. Therefore, we performed therapeutic penetrating keratoplasty using a preserved corneal button (Figure 3b).

### 2.2. Surgical Procedure and Postoperative Course

Under general anesthesia, the affected corneal tissue of the patient was circularly removed with a diameter of 7.25 mm. A corneal button (7.5 mm in diameter) of the preserved cornea was prepared. The button of the donor was sutured at 16 points with 10-0 nylon strings. The degenerated lenticular structure and anterior vitreous were removed through the corneal hole after removal of the lesion using a Constellation^®^ (Alcon, Fort Worth, TX, USA) microsurgical system with the open sky technique. After the surgery, natamycin ointment was applied and the cornea was covered with a soft contact lens. The degree of pain reduced one day after surgery (Figure 4a). The same antifungal medication used before surgery was continued after operation. Moreover, systemic prednisolone 30 mg/day was started. Three weeks after operation, the visual acuity was hand motion (Figure 4b).

### 2.3. Histopathology of the Affected Corneal Tissue

The excised cornea was prepared for histopathological examination using hematoxylin and eosin (HE) staining and Grocott + light green staining for fungi detection. Inflammation and fungus invasion were observed within the stroma collagen lamellae throughout the full thickness of the stroma around the ulcer perforation and in the anterior stroma apart from the ulceration (Figure 5).

## 3. Discussion

Fungal keratitis reportedly occurs in patients with previous injuries and local or systemic immune-compromised conditions. In the present case, the patient received pterygium excision and then developed a corneal ulceration. Since the primary doctor considered that there was a possibility this corneal ulceration could be related to rheumatoid arthritis, topical glucocorticoid was administered.

Infection of the tissue by *Colletotrichum* spp. is usually related to a preexisting wound. Although *Colletotrichum gloeosporioide* infection is usually limited to the anterior stroma with less severity [2,3,4,9,10], there were reports of corneal infection from this fungus in surgically treated patients with local or systemic compromised backgrounds [6,7,11,12]. In the current study, we observed *Colletotrichum*
*gloeosporioides* invasion of the stromal tissue due to keratitis. The infection is usually treated by medications, and therefore it is uncommon to observe histopathology of the affected cornea.

We here report a case of corneal stromal perforation caused by this fungus. Possible causes the lesion becoming worse in this case include preexisting surgical trauma, ulceration, and topical glucocorticoid administration. Glucocorticoid application could suppress stromal infection, even though it did not exert a therapeutic effect on a fungal infection. This case suggests that topical glucocorticoid administration before ruling out a possibility of infection potentially worsens the lesion, even though the original pathogen is less virulent.

## Figures and Tables

**Figure 1 pathogens-11-00526-f001:**
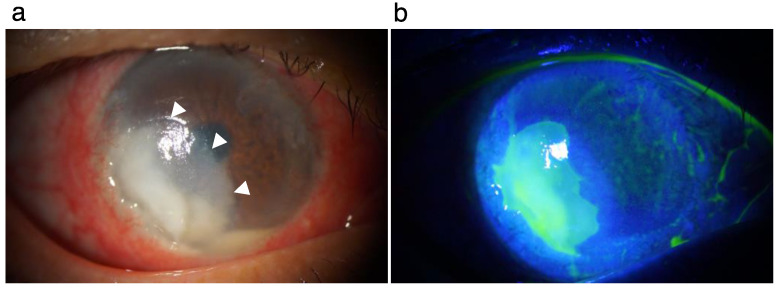
Corneal lesion at the first consultation. (**a**). Severe hyperemia, stromal infiltration (arrowheads), epithelial defect, and hypopyon are shown. (**b**). Fluorescein staining defines the area of the epithelial defect.

**Figure 2 pathogens-11-00526-f002:**
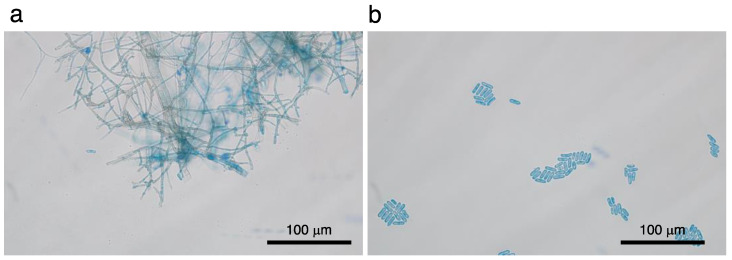
Detection of *Colletotrichum gloeosporioides*. (**a**). Filamentous fungus is detected from the lesion. (**b**). Biological culture allows the growth of *Colletotrichum gloeosporioides*. Bar, 100 μm (**a**,**b**).

**Figure 3 pathogens-11-00526-f003:**
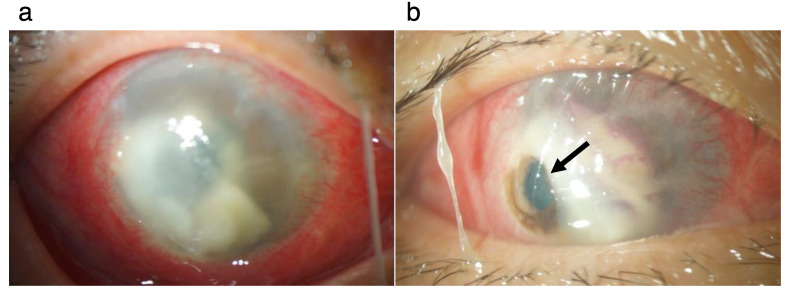
Findings in the anterior segment before therapeutic penetrating keratoplasty. (**a**). At 4 days after the first consultation, the lesion in the cornea is enlarged compared with that at the first consultation shown in Figure 1. (**b**). Perforation (arrow) of the cornea is observed in the inferotemporal area.

**Figure 4 pathogens-11-00526-f004:**
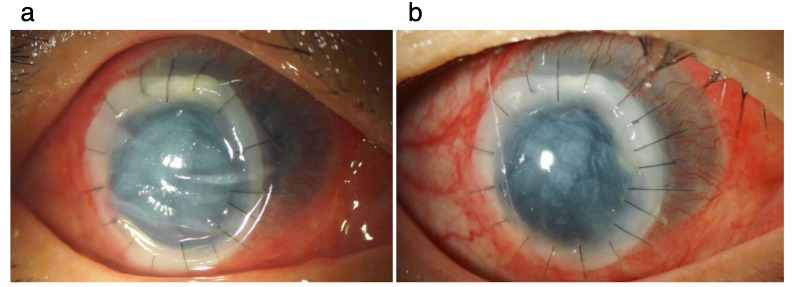
Findings after therapeutic penetrating keratoplasty at day 1 and 3 weeks. The graft is well suture-fixated to the affected cornea, although stromal edema is observed at day 1 after operation. At week 3, stromal edema of the graft seems to be reduced.

**Figure 5 pathogens-11-00526-f005:**
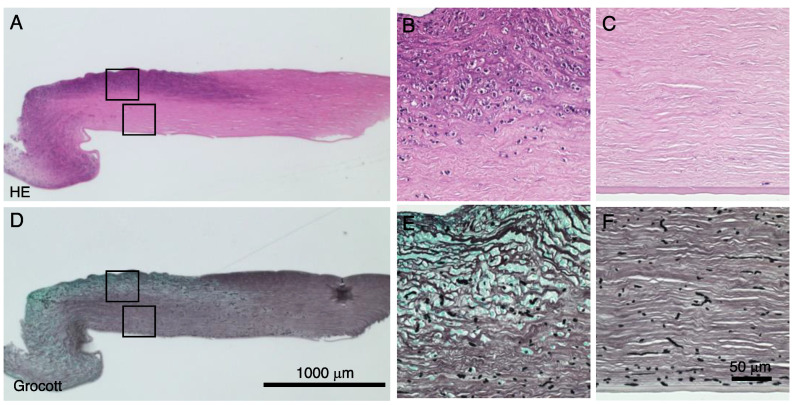
Histopathology of the affected corneal tissue. Hematoxylin and eosin (HE) staining (**A**–**C**) and Grocott + light green staining (**D**–**F**) for fungi detection are shown. Inflammation and fungus invasion were observed throughout the full thickness of the stroma around the ulcer perforation and in the anterior stroma apart from the ulceration (**A**,**D**). Higher magnification observation shows inflammatory cells and that fungi are invading the collagen fibers of the stroma. Bar, 100 μm (**A**,**D**); 50 μm (**B**,**C**,**E**,**F**).

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
