# Peer review of "Therapeutic Penetrating Keratoplasty in a Case of Corneal Perforation Caused by Colletotrichum gloeosporioides Infection"

_pathogens, 2022, doi:10.3390/pathogens11050526_

Round 1

Reviewer 1 Report

Authors described very interesting case of a 73 year-old woman with infection of Colletotrichum gloeosporioides. The case is very well documented and methods are properly used.

Authors should add Institutional Review Board Statement.

Minor spelling errors are also present.

Author Response

Authors’ response to comments and suggestions

Reviewer 1

Comment: Authors should add Institutional Review Board Statement.

Response: Yes, thank you very much. The Institutional Review Board of Wakayama Medical University dose not require an approval for that a report of a single case. We described this in the revise text.

Comment: Minor spelling errors are also present.

Response: Thank you very much. We have corrected them.

Reviewer 2 Report

i.e - such as in lieu of i.e

Line 44 - I am uncertain what biological examinations refers to. What was tested (eg on Sabouraud’s agar)

I note that the patient was started on Natamycin 5%. Shiraishi et al (doi: 10.1089/jop.2011.001) have published a series of 3 cases with sensitivity testing demonstrating relative resistance to flucytosine, fluconazole, and natamycin. Can the authors kindly provide information pertaining to the drug of choice, sensitivity testing if present, and whether this might have resulted in clinical deterioration as described?

The authors have alluded to use of topical corticosteroids in the setting of a patient with a known history of rheumatoid arthritis for what I presume was PUK.No information has been provided regarding the type of corticosteroid (potency), frequency of administration, when this was administered and ceased. What also prompted the patient’s eye specialist to change course to clinch the diagnosis of fungal keratitis instead. 

There have been cases of Colletotrichum related infective keratitis already described and published in the literature. It is also well established that topical corticosteroids can worsen infective keratitis should it not be given in isolation for bacterial keratitis and caution had with fungal or parasitic causes. What would the novel slant from the authors be in this instance? 

Author Response

Authors’ response to comments and suggestions

Reviewer 2

Comment: English language and style are fine/minor spell check required

Response: English writing has been improved.

“i.e “ has been replaced by “such as “.

Comment: Line 44 - I am uncertain what biological examinations refers to. What was tested (eg on Sabouraud’s agar)

Response: Thank you very much. We used culture by Sabouraud’s agar.

Comment: I note that the patient was started on Natamycin 5%. Shiraishi et al (doi: 10.1089/jop.2011.001) have published a series of 3 cases with sensitivity testing demonstrating relative resistance to flucytosine, fluconazole, and natamycin. Can the authors kindly provide information pertaining to the drug of choice, sensitivity testing if present, and whether this might have resulted in clinical deterioration as described?

Response: Thank you very much. Unfortunately, it was difficult to perform drug-sensitivity testing because the volume of the fungi was small and was not enough to perform such testing. We therefore administered generally approved anti-fungus drug. Nevertheless, we considered the major cause of the deterioration included the depth of the fungus invasion in the deeper stroma and pre-administration of glucocorticoid. This point has been incorporated in the revised manuscript.

Comment: The authors have alluded to use of topical corticosteroids in the setting of a patient with a known history of rheumatoid arthritis for what I presume was PUK. No information has been provided regarding the type of corticosteroid (potency), frequency of administration, when this was administered and ceased. What also prompted the patient’s eye specialist to change course to clinch the diagnosis of fungal keratitis instead. 

Response: Thank you very much. The patient received topical administration of 0.1% levofloxacin and 0.1% betamethasone sodium phosphate for one month upon diagnosis of peripheral ulcerative keratitis (PUK) before consultation with us. At the consultation with us, we discontinued betamethasone application. In our inpatient clinic, besides topical levofloxacin, the patient received infusion administration of Cefmenoxime Hydrochloride and A mixture infusion drug of Ampicillin Sodium (4 g) and Sulbactam Sodium (2 g) per day until we received biological culture test. Then, we ran fungal culture test, because the above topical/systemic antibiotics treatment did not seem therapeutically effective.

Comment: There have been cases of Colletotrichum related infective keratitis already described and published in the literature. It is also well established that topical corticosteroids can worsen infective keratitis should it not be given in isolation for bacterial keratitis and caution had with fungal or parasitic causes. What would the novel slant from the authors be in this instance

Response: Thank you very much for a useful suggestion to brush up the discussion. The points the authors would like to report here include; It is uncommon that (1) Colletotrichum spp. caused severe corneal tissue destruction, that needed to be treated by penetrating keratoplasty, and (2) Here, we had an opportunity to report the way of pathogen invasion in the stromal tissue of Colletotrichum gloeosporioides keratitis. It is usually treated by medications and thus it is uncommon to have a chance to observe histopathology of the affected cornea.

Round 2

Reviewer 2 Report

I thank the authors for revising the manuscript and for inclusion of further relevant clinical information. I am interested in the clinical rationale of commencing such a broad spectrum of systemic antibiotics in view of the patient's clinical presentation. While systemic antibiotics are rightly started in the presence of endophthalmitis, or scleral involvement (as evident from the clinical photos, there is very little data to support use of systemic penicillins, for example, in the treatment of ocular conditions due to concerns about achieving therapeutic levels). Would be grateful if the authors could provide clinical rationale for each of the systemic medications administered in this setting. 

I would suggest the authors examine the manuscript to ensure that formatting remains consistent (every one hour vs every 4 hours for instance). Would suggest inclusion of the route of administration of antifungals (topical presumably)

I am interested in how the authors are able to accurately determine that it was purely fibrin in the anterior chamber. In the setting of an infection, it is more likely to be a hypopyon. Fibrinous collections are more typically used in the description of HLAB27 type AAU or Behcets disease for instance. 

Author Response

Authors’ comment to the reviewer:

We newly added sentences to introduce the patient in Lines 41 – 48, that is summarized in the abstract, but had not shown in the text.

The points revised upon 2nd revision are marked by markers of green or magenta.

Authors’ response to the reviewer’s comments:

Comment: I thank the authors for revising the manuscript and for inclusion of further relevant clinical information. I am interested in the clinical rationale of commencing such a broad spectrum of systemic antibiotics in view of the patient's clinical presentation. While systemic antibiotics are rightly started in the presence of endophthalmitis, or scleral involvement (as evident from the clinical photos, there is very little data to support use of systemic penicillins, for example, in the treatment of ocular conditions due to concerns about achieving therapeutic levels). Would be grateful if the authors could provide clinical rationale for each of the systemic medications administered in this setting.

Response: Thank you very much for the important comment. The guideline for the treatment of the severe corneal infectious diseases established by the Japanese Ophthalmological Society recommends systemic infusion administration of cephem antibiotics. In the present case, we considered that the case was severe with hypopyon 1 month after the onset and followed this guideline. We described this in the 2nd revised version.

Comment: I would suggest the authors examine the manuscript to ensure that formatting remains consistent (every one hour vs every 4 hours for instance). Would suggest inclusion of the route of administration of antifungals (topical presumably)

Response: Thank you very much. We corrected such points.

Comment: I am interested in how the authors are able to accurately determine that it was purely fibrin in the anterior chamber. In the setting of an infection, it is more likely to be a hypopyon. Fibrinous collections are more typically used in the description of HLAB27 type AAU or Behcet’s disease for instance.

Response: Thank you very much. We observed hypopyon at the first consultation to us, that was associated with a niveau formation. AS the reviewer pointed out the fibrin reaction observed through slit lamp was in the anterior chamber besides hypopyon.
